# Multidimensional Motivational Climate Questionnaire in Physical Education at the Situational Level of Generality (MUMOC-PES)

**DOI:** 10.3390/ijerph20054202

**Published:** 2023-02-27

**Authors:** Omiros Vlachos, Athanasios G. Papaioannou

**Affiliations:** Department of Physical Education and Sport Science, University of Thessaly, 42100 Trikala, Greece

**Keywords:** empowering, disempowering, self-determination, achievement goals, satisfaction

## Abstract

The main aim of this study was to investigate the construct validity of a new MUltidimensional MOtivational Climate questionnaire in Physical Education (PE) at the Situational level of generality (MUMOC-PES), capturing four dimensions of empowering (autonomy support, task-involvement, relatedness support, structure) and three dimensions of disempowering (controlling, relatedness thwarting, ego-involvement) climate. Nine hundred and fifty-six adolescent students completed the new measure alongside measures of mastery and performance approach/avoidance climate and satisfaction. Confirmatory factor analysis supported the construct validity of the MUMOC-PES. Student satisfaction in PE corresponded positively to empowering and negatively to disempowering climate dimensions. Controlling for age, gender and within-class individual differences in perceived empowering and disempowering dimensions, class average scores on perceived empowering climate had significant effects on student satisfaction, implying predictive validity for the MUMOC-PES. Structural Equation Modeling (SEM) suggested that perceived autonomy support and relatedness thwarting had direct positive and negative effects on satisfaction respectively. Moreover, effects of perceived structure and thwarting relationships on satisfaction were mediated through a mastery climate construct capturing the linkage between perception and mastery goal. The results are discussed in relation to existing measures and literature on motivational climate and the future use of MUMOC-PES in research and PE teachers’ training.

## 1. Introduction

Physical Education (PE) teachers have a crucial role in creating a school motivational environment that promotes adaptive motivational outcomes such as cooperation, choice and enjoyment in PE [1,2]. Duda [3] conceptualized motivational climate as empowering and disempowering based on a combination of two popular theories of motivation, Self-Determination Theory (SDT; [4]) and Achievement Goal Theory (AGT; [5,6,7]). Accordingly, Duda and her colleagues [3] developed a measure to capture the perceived Empowering-Disempowering Climate (EDC; [8]).

Here we present another measure of perceived EDC, which captures a more extended spectrum of EDC, capturing seven climate dimensions, including thwarting relationships and structure that have not been captured in the past [8], although they are considered important for student motivation and enjoyment in PE [9,10]. Importantly, while trying to establish this measure’s content validity, we aimed at its future concurrent use with the Multidimensional Motivational Climate Observation System (MMCOS) [11] in studies adopting triangulation research methods of EDC measurement. Moreover, we tested theoretical assumptions about the location of the present measure alongside instruments of perceived motivational climate in PE that comply with the rule of compatibility [12], presenting the strengths and weaknesses of all of them. Finally, we examined whether within and between-class differences in perceived EDC accounts for significant variance of student satisfaction in PE, essentially also testing predictive validity for the new measure. 

## 2. Links between SDT and AGT in the Measurement of Motivational Climate

Researchers have adopted Self-Determination Theory (SDT) and AGT to describe and capture the motivational climate in PE and sport settings. In SDT, significant others like teachers, through their behaviors can influence the type of pupil motivation by supporting or thwarting the basic psychological needs for competence (being effective in contexts), autonomy (to have a choice and act with their own decision) and relatedness (being with and receive care from other people) [13]. In addition, class climate is considered important because it fosters students’ intrinsic motivation and motivational outcomes such as satisfaction, personal development, and well-being [13,14]. Furthermore, teachers maintaining autonomy and structure in their class exhibit positive teaching behaviors, job satisfaction, and contribute to students’ learning [15,16,17].

Achievement Goal Theory (AGT; [6,7]) explains how individuals conceptualize or demonstrate ability and define success through subjective/task-involving or normative/ego-involving criteria. Ames [5] proposed that teacher’s behaviors and practices could create a climate promoting different achievement goals that might affect differently students’ learning, involvement and other achievement-related behaviors. When students perceive that their class environment emphasizes learning and personal improvement, they feel competent, satisfied and intrinsically motivated [18]. On the other hand, ego-involving climate emphasizes normative performance, overcoming others and normative ability [5,19]. An ego-involving climate has negative effects on student motivation when teachers promote intra-class rivalry, praise the most skillful, whereas the low potential pupils exhibit discomfort and boredom [20,21]. 

Elliot and Church [22] proposed an hierarchical approach and avoidance achievement motivation model by distinguishing performance goals into performance-approach and performance-avoidance goals. In performance-approach goals, individuals focus on expressing competence relative to others, while mastery goals focus on developing competence or task mastery. Performance-avoidance goals are linked to fear of negative evaluation of competence from others [22]. Accordingly, Papaioannou and colleagues [12] developed a measure capturing perceptions of teacher behaviors promoting performance-approach and performance-avoidance goals, alongside mastery goals in PE classes. Their research findings showed that perceptions of mastery-oriented climates positively relate to student satisfaction in PE [23].

There are several linkages between AGT and SDT constructs as well as between these constructs and student motivational outcomes, including satisfaction in PE. For example, Standage et al. [24] found that a task-involving climate predicted autonomy, competence, and satisfaction. Furthermore, research in PE settings has shown that a task-involving and autonomy-supportive motivational climate could enhance intrinsic motivation, satisfaction, and willingness to participate in physical activity or school settings [20,25]. In addition, when PE teachers support student needs of autonomy, competence, and relatedness, they promote an autonomy-supportive motivational climate, contributing to their pupils’ positive outcomes and satisfaction [26]. 

García-González et al. [27] and Chen et al. [28] found that student perceptions of autonomy support and task-involving climate were positively related to the satisfaction of their basic psychological needs. García-González et al. [27] also reported that students’ perceived task-involving climate was significantly connected with basic phycological needs satisfaction. On the contrary, an ego-involving climate was positively connected with basic psychological needs frustration, leading to incompetence and lack of motivation [27]. These pieces of evidence revealed the linkages between the AG and SD theories, suggesting that task-involvement help individuals to express their autonomous behavior, because when people are task-involved, their motivation to accomplish a task comes from intrinsic criteria, in contrast to ego-involvement, which thwart intrinsic concerns [29,30]. 

Previous studies revealed that task-involvement is connected with intrinsic motivation, enjoyment, and less boredom [24,25]. Jaakkola et al. [23] confirmed that students who perceived the motivational climate as highly task-involving were more autonomous regulated, less motivated and enjoyed more the PE lesson than those students who felt their climate less task-involving. Moreover, the negative relationship of satisfaction and performance approach and avoidance goals implies that performance climate negatively affects pupil perceptions of enjoyment [23]. Additionally, while high perceived competence leads individuals toward success and performance-approach goal adoption, low perceived competence orients people to avoid failure and endorse performance-avoidance goals [22,31]. 

## 3. Measurement of Motivational Climate in PE

Based on AGT, Papaioannou [32] developed the *Learning and Performance Orientations in Physical Education Classes Questionnaire* (LAPOPEQ) to measure student perceptions of achievement orientations in PE classes. This instrument captures two dimensions of task-involving climate (teacher-initiated learning orientation and student learning orientation) and three dimensions of ego-involving climate (students’ competitive orientation, students’ worries about mistakes and outcome without effort) [32]. Later Papaioannou et al. [12,33] developed the *Physical Education Teacher’s Emphasis on Goals questionnaire* (PETEG) at the contextual and situational levels of generality. Of particular interest in this study is the situational level implying measurement of student perceptions of the PE climate at a particular time point/day, which is affected by the specific teacher behaviors and task structure on that day [12]. These scales were based on the rule of compatibility, implying that measures of perceived motivational climate have the strongest association with motivational outcomes when perceptions of teachers’ emphasis on a particular goal are directly linked with the corresponding students’ goal adoption [12] (p. 42). These 4-factor measures captured teachers’ behaviors promoting four goal orientations in PE: mastery, performance-approach, performance-avoidance, and social approval.

The first effort to combine AGT and SDT in the measurement of motivational climate in PE was made by Goudas and Biddle [34]. Based on LAPOPEQ they developed the *Physical Education Class Climate Scale* (PECCS). To the LAPOPEQ factors, they added two SDT factors (student perception of choice and teacher support) and excluded one factor from LAPOPEQ subscales (outcome without effort). Similarly, Soini et al. [35] developed a motivational climate in PE questionnaire capturing task-involving, ego-involving, autonomy support and social relatedness. 

Duda [3] proposed the terms *Empowering* and *Disempowering* to describe a motivational climate in sport integrating aspects from AGT and SDT. An *Empowering* environment promotes task-involving, autonomy support, and social support. On the other hand, a *Disempowering* environment emphasizes ego-involving, controlling and thwarting relationships in sport. 

Milton et al. [8] developed the *Empowering and Disempowering Motivational Climate Questionnaire in Physical Education* (EDMCQ-PE) consisted of three empowering scales (task-involving, autonomy, social support) and two disempowering scales (ego-involving and controlling). The latter did not include a scale assessing thwarting social relationships that is considered essential in contemporary SDT research [10,36]. Moreover, the EDMCQ-PE does not assess “structure” which is captured in observational measures of EDC [11,37] and is considered important [38,39] because it keeps the guidance of the PE teachers’ and students’ goals [40,41]. In a typical PE context, a well-structured class providing clear direction and support during the classes can increase motivation, enjoyment, and pupils’ sense of purpose [13,42]. Research showed positive correlations between motivation and PE satisfaction with structured PE [9]. 

Milton and colleagues [8] adapted items from existing sport climate measures for their PE climate measure, nevertheless, some EDMCQ-PE items are not very pertinent to the PE content that differs from sport in terms of aims, management/organization and task structures. For example, their item “*my teacher made sure everyone had an important role in the class*” might not be a strong indication of a climate promoting mastery goals in PE. We are not aware of existing studies investigating the concurrent validity of this climate measure in relation to previous climate measures, neither the multilevel effects of responses to the EDMCQ-PE on motivational outcomes in PE. 

## 4. The Present Study

This study draws from AGT, SDT and Duda’s [3] conceptualization of empowering and disempowering motivational climate, aiming to develop the *MUltidimensional MOtivational Climate questionnaire in Physical Education at the Situational level of generality* (MUMOC-PES). In comparison to Milton et al.’s [8] instrument, an advancement of MUMOC-PES is its 7-factor structure, capturing four empowering dimensions (task-involving, autonomy support, relatedness support, structure) and three disempowering dimensions (ego-involving, controlling, relatedness thwarting) in line with observational measures of EDC [8]. Two experts in student motivation in PE were asked to investigate existing instruments in PE and then to develop items for a new 7-factor EDC measure in PE at the situational level of generality (i.e., capturing PE climate at a particular time point/daily class) [43]. 

An additional novelty of the MUMOC-PES in relation to previous climate measures was the instruction to the two experts to develop items that would capture student perceptions of empowering-disempowering PE climate in the same way as external observers/motivation specialists would perceive the same PE class. We considered this an important feature of motivational climate measures in studies adopting triangulation methods, combining students’ perceived climate in conjunction with motivational climate assessed by external observers. Indeed, existing studies revealed non-significant or very low relationships between the MMCOS and perceived climate measures that have not been developed for concurrent use with the MMCOS [44]. Furthermore, MUMOC-PES might be particularly important in intervention studies targeting perceived motivational climate and concurrently testing the fidelity of manipulating motivational climate. Finally, the MUMOC-PES might be used in teacher training because it might provide immediate student feedback to teachers. Accordingly, the two experts also investigated observational measures, particularly the MMCOS [11,37], which was the most relevant to the aims of this study. 

Despite these strengths, this approach might have also a restraint. The development of these new items cannot comply with Papaioannou et al.’s [12,33] suggestion that pupils’ perceived climate should be strongly associated with the corresponding pupils’ goal adoption, because external observers of the PE climate cannot record pupil goal adoption and the corresponding student perception-goal linkage. Accordingly, when we focus at the individual; it is expected that students’ motivational outcomes would be more proximal to mastery and performance climate scales of the PETEG in relation to the task and ego-involving climate scales of the MUMOC-PES. An initial attempt to test this assumption was made in this study. 

To help students complete the entire battery of questionnaires within short time, just after their participation in a PE class and before moving to other school classes, we examined one motivational outcome, satisfaction in PE.

The purpose of the present study was threefold, that is, to examine: (a)The construct validity of the new MUMOC-PES. Beyond CFA, internal convergent and divergent validity was examined through bivariate relationships between the subscales of the MUMOC-PES. In addition, concurrent validity was also examined by investigating the relationship of the MUMOC-PES subscales with the subscales of the PETEG at the Situational Level of Generality [12].(b)The assumption that some of the effects of MUMOC-PES is more distal to motivational outcomes in relation to climate measures capturing students’ perception-goal linkage (PETEG).(c)The multilevel effects of perceived motivational climate on pupil satisfaction in PE. Significant level 2 effects indicating effects of class means on empowering or disempowering climate on satisfaction would offer evidence of predictive validity for the MUMOC-PES.

### Research Hypotheses

**H_1_.** *CFA will confirm the 7-factor structure of the MUMOC-PES*.

**H_2_.** *Pearson correlations among the MUMOC-PES subscales will provide evidence of internal convergent and divergent validity for the MUMOC-PES, while correlations between PETEG and MUMOC-PES subscales will provide evidence of concurrent validity for the MUMOC-PES*.

**H_3_.** *At the individual level of analysis, most effects of the MUMOC-PES on student satisfaction will be mediated through the PETEG, particularly through PETEG mastery*. 

**H_4_.** *The empowering (support of autonomy, relatedness, task-involvement, and structure) will have multilevel positive effects on pupil satisfaction in PE, suggesting that both within class and between class variation in empowering climate perceptions of empowering climate has an impact on student satisfaction in PE*.

## 5. Method

### 5.1. Participants

Participants were nine hundred fifty-six pupils (*n =* 424 boys and *n =* 532 girls) in 53 coeducational classes from two regions in central Greece. Their age ranged from 11 to 12 (*n =* 101 boys, *n =* 130 girls, in primary school); 13 to 15 (*n =* 239 boys, *n =* 295 girls, in junior high school) and 16 to 18 *(n =* 84 boys, *n =* 107 girls, in senior high school). The average number of pupils per class was 18.04 (*SD* = 5.27) with no class including less than 13 students except of four classes numbering 9–12 pupils each of them.

### 5.2. Procedure

This study was conducted with the approval of the Ministry of Education and the Ethics Committee of the researchers’ university. All parents provided informed consent and student participation was voluntary. The study was conducted during a daily PE lesson. Before the study’s commencement, a researcher informed the headmasters, PE teachers, pupils, and parents about the research design. Teachers were asked to adapt their daily teaching in a 30-min session instead of the standard 45-min lesson. At the end of the teaching session, in the remaining 15 min, pupils completed the anonymous instruments shown below under the researcher’s presence but without the teacher’s presence. 

### 5.3. Measures

*Motivational Climate*. To assess the perceived motivational climate at the situational level, we used both the MUMOC-PES and the PETEG. The development of the MUMOC-PES was based on the modification of items used by external observers while using the MMCOS [11,37]. The MUMOC-PES consisted of seven factors (autonomy support, controlling, task-involving, ego-involving, relatedness support, relatedness thwarting and structure) and 38 items capturing PE teachers’ “*behavioral strategies*”. The responses are given on a 4-point Likert scale ranging from 0 (*none*) to 3 (*gave great emphasis*). Following the stem “*In the current physical education class, PE teacher is …*”, participants responded to the items reflecting autonomy support (*six items; e.g., … is providing meaningful choices*), controlling (*seven items; e.g., … using extrinsic rewards*), task-involving (*five items; e.g., … is recognizing effort and improvement*), ego-involving (*five items; e.g., … is emphasizing inferior or superior ability*), relatedness support (*five items; e.g., … is adopting a warm communication style*), relatedness thwarting (*six items; e.g., … is showing a lack of care and concern for players*) and structure (*four items; e.g., … is providing guidance throughout exercises*). 

The PETEG [12] has been previously used in this country to examine student perceptions about the motivational climate created by the PE teacher at a situational level of generality. Here we used three scales of the PETEG (mastery, performance-approach, performance-avoidance). Under the stem “*In today’s physical education class, our PE teacher …*”, participants answered to the items reflecting mastery (*six items; e.g., … was very satisfied when someone showed improvement after hard effort*), performance-avoidance (*six items; e.g., … made me worry if they say that I am not capable*) and performance-approach climate (six *items; e.g., … was pleased with students showing that they were more capable than others*) on a 5-point Likert scale from 1 (*Strongly Disagree*) to 5 (*Strongly Agree*). 

*Satisfaction in PE*. An adapted version of *Classroom Satisfaction/Enjoyment Scale* (Duda and Nicholls, 1992) [45] was used to measure pupil satisfaction in PE lesson. Examples of this five-item scale are: “*During Physical Education class today… I had fun*” or “*… I found it interesting*”. Participants responded on a 5-point Likert scale ranging from 1 (*Strongly Disagree*) to 5 (*Strongly Agree*). A translated version of this questionnaire has been previously used in this country with good psychometric properties [12,33]. 

### 5.4. Statistical Analysis

Initially, a Confirmatory Factor Analysis (CFA) for all three motivational climate scales was conducted. Concerning MUMOC-PES, seven different factors (autonomy support, controlling, task-involving, ego-involving, relatedness support, relatedness thwarting and structure) were constructed. A 3-factor model (mastery, performance-approach, performance-avoidance) was structured to test the PETEG, and the one-factor model was tested for satisfaction. Across all models, no correlation of residuals was allowed and the Maximum Likelihood (ML) estimation method was selected. The following model Goodness of Fit Indices (GFIs) were considered to evaluate how well the data fit to this model: chi-square (*χ*^2^), the Tucker–Lewis Index (TLI), Comparative Fit Index (CFI), Root Mean Square Error of Approximation (RMSEA). Moreover, Browne and Cudeck’s [46] Expected Cross-Validation Index (ECVI) along with a 95% confidence interval was used. The index of *χ*^2^ (chi-square) is affected by the sample size, while the TLI is not [47]. The TLI and CFI indices should have values as close to 1 (or >0.95). The RMSEA values below 0.06 indicate a good model adjustment [48,49]. Following this, descriptive statistics, Pearson’s *r* correlations among all scale scores were conducted. Moreover, between-classes differences in all scale scores were examined and related ε^2^ effect sizes were computed that are considered stricter than η^2^ and less biased than η^2^ and ω^2^ [50]. 

At the individual level of analysis Structural Equation Modelling (SEM) with ML estimation was performed using IBM^®^ SPSS^®^ AMOS 22 software (IBM, Armonk, New York, NY, USA) to examine the effects of MUMOC-PES factors on satisfaction in PE and the mediational role of PETEG. Models are described in the results section.

Two MUMOC-PES climate dimensions were computed, empowering (average of autonomy support, task-involving, relatedness support, and structure scale scores) and disempowering (average of controlling, ego-involving, and relatedness thwarting). To investigate the effects of within-class and between-class variation in empowering and disempowering climate on pupil satisfaction, multilevel regression models were run through Mixed Models of IBM^®^ SPSS^®^ [51] using pupil satisfaction as Dependent Variable (DV). The Intra-Class Correlation for this DV was 0.11 well above the 0.05 value that is considered prima facie evidence of a group effect that should be investigated through multilevel modelling [52]. Three level 1 predictors were used, perceived empowering and disempowering climate dimensions centered on class mean for each climate dimension score respectively and gender. Three level 2 predictors were used, the class mean for empowering and disempowering score respectively and age. 

## 6. Results

### 6.1. Confirmatory Factor Analysis (CFA) and Reliability

A Confirmatory Factor Analysis (CFA) was implemented to examine the construct validity of the adapted questionnaire MUMOC-PES. The CFA’s first results showed an insufficient fit for the initial scale with the 38 items (*χ*^2^ = 1396.856, *df* = 443, *TLI* = 0.863, *CFI* = 0.878, *RMSEA* = 0.052 (90% CI = 0.049–0.055, *ECVI* = 2.057). After removing six items, from autonomy (1 item), controlling (3 items), relatedness support (2 items), task involving (1 item), ego involving (2 items), and structure (1 item) due to low factor loadings, the goodness-of-fit indices were satisfactory for the seven dimensions: *χ*^2^ = 821, *df* = 278, *TLI* = 0.923, *CFI* = 0.934, *RMSEA* = 0.045 (90% CI = 0.042–0.049), *ECVI =* 1.013. Loadings ranged from 0.49 to 0.77, Table 1. These findings support the 7-factor structure of the MUMOC-PES (H_1_). 

A CFA was conducted to examine the construct validity of the PETEG. Initial results showed an insufficient model fit, *χ*^2^ = 626.294, *df* = 132, *TLI* = 0.837, *CFI* = 0.859, *RMSEA* = 0.063 (90% Cl = 0.058–0.068), *ECVI* = 0.737. After removing four items from performance-approach (2 items) and performance-avoidance (2 items) subscales, due to low factor loadings, the factorial validity was satisfactory for the remaining items: *χ^2^* = 225.744, *df* = 74, *TLI* = 0.929, *CFI* = 0.942, *RMSEA* = 0.046 (90% CI = 0.040–0.053), *ECVI* = 0.301. 

A third CFA for the satisfaction in PE scale revealed unsatisfactory GFIs for a model consisted of one latent and five observed variables, *χ^2^* = 101.854, *df* = 5, *TLI* = 0.908, *CFI* = 0.954, *RMSEA* = 0.142 (90% CI = 0.119–0.167), *ECVI* = 0.128. After removing one item, the goodness-of-fit indices were satisfactory: *χ^2^* = 15.518, *df* = 2, *TLI* = 0.974, *CFI* = 0.991, *RMSEA* = 0.084 (90% CI = 0.098–0.125), *ECVI* = 0.033. 

The reliability alpha coefficients for the scales comprising the remaining items of each aforementioned factor were mostly acceptable (0.56 to 0.84). In addition, McDonald’s Omega (ω) coefficient was computed (0.56 to 0.84) because researchers suggested that it has some advantages over alpha and it can be used as an alternative of the scales’ internal consistency [53]. The low reliability value of the ego-involving and controlling scales was due to the small number of the items after item removal [54].

### 6.2. Correlation Analysis

Pearson correlations were conducted to investigate H_2_. As shown in Table 2, high or moderate positive correlations emerged among MUMOC-PES scale scores assessing perceived autonomy support, task-involving climate, relatedness support, and structure. In contrast, these four empowering climate scale scores had negative relations with perceptions of controlling, ego-involving, and relatedness thwarting climate scores. On the other hand, moderate positive correlations emerged among perceived ego-involving, controlling and thwarting relationships scale scores of MUMOC-PES, which are assumed to assess disempowering climate. 

The MUMOC-PES perceived empowering scale scores (autonomy support, task-involving, relatedness support, and structure) were positively associated with perceived mastery and negatively associated with perceived performance approach and performance avoidance scale scores of the PETEG. On the other hand, the MUMOC-PES perceived disempowering dimensions (controlling, ego-involving, relatedness thwarting) were positively related to perceived performance-approach and performance-avoidance climate and negatively related to perceived mastery climate assessed by PETEG. 

Gender was negatively related to satisfaction and perceived controlling, ego-involving, relatedness thwarting, performance avoidance, suggesting higher scores in these variables for males than females. Age was positively related to perceived thwarting relationships. On the other hand, age was negatively related to satisfaction and perceptions of mastery, task-involving, structure, relatedness support and controlling climate Table 2. 

Based on these findings, two composite climate scores were created: (1) empowering, indicating the average in autonomy support, task-involving, relatedness support, and structure scale scores, and (2) disempowering, denoting the average in controlling, ego-involving, and relatedness thwarting scale scores. 

Satisfaction in PE had positive relationships with all “*Empowering*” dimensions (i.e., autonomy support, task-involving, relatedness support, and structure) and perceived mastery climate, and negative relationships with all “*Disempowering*” dimensions (i.e., controlling, ego-involving and relatedness thwarting) and perceived performance approach and avoidance climate. 

Means, standard deviations, Cronbach’s alpha (*α*), McDonalds Omega (ω), ε^2^ and correlation analysis are presented in Table 2.

### 6.3. Differences between Teachers

Significant (*p* < 0.001) between-classes differences emerged across all scale scores. Effect sizes imply large between-classes differences across all climate dimensions of the MUMOC-PES (ε^2^ > 0.15) and most dimensions of the PETEG (ε^2^ > 0.10) in Table 2.

### 6.4. SEM

An initial SEM model was constructed including age, gender, seven latent variables reflecting each of the seven climate dimensions of the MUMOC-PES and the latent variable satisfaction. It was assumed that each of the seven climate dimensions, as well as age and gender, would have direct effects on satisfaction. This model revealed that three latent variables, autonomy, structure, and thwarting relationship, as well as age and gender, had significant effects on latent variable satisfaction. After exclusion of the remaining four latent variables with no significant effects on satisfaction, the standardized beta weights indicating direct effects on satisfaction were as follows: autonomy (β = 0.20, *p* < 0.01), structure (β = 0.25, *p* < 0.01) and thwarting relationships (β = −0.22, *p* < 0.001) (GFIs for this model, χ^2^ = 414, *df* = 143, TLI = 0.942, CFI = 0.952, RMSEA = 0.045).

Then, a model investigating the mediating role of perceived mastery climate assessed through PETEG was constructed. Initially it was assumed that perceived autonomy, structure, and thwarting relationships would have both direct and indirect effects on satisfaction through perceptions of mastery. As shown in Figure 1, after retaining only the paths with significant effects, controlling for age and gender, the effect of structure on satisfaction was fully mediated through perceived mastery; autonomy had only direct effect on satisfaction, while the negative effects of thwarting relationships were both direct and indirect through perceptions of mastery. 

### 6.5. Multilevel Modelling

Fixed effects from multilevel regression analysis are shown in Table 3. These findings imply that controlling for gender and age, both within and between class variation in perceived empowering score had positive effects on satisfaction. On the other hand, while within-class variation in perceived disempowering score had negative effects on satisfaction, between-class variation in perceived disempowering climate was not associated with satisfaction. 

## 7. Discussion

The main aim of this study was to establish construct validity of a new EDC questionnaire in PE at situational level (MUMOC-PES). The measure was developed to capture all EDC dimensions suggested by Duda [3], including thwarting relationships and structure suggested by Smith et al. [11]. Moreover, all items were constructed to capture empowering-disempowering climate in the same way as external observers would assess it at a specific point of time (situational level of generality).

The CFA and reliability analysis results supported the first hypothesis showing that the MUMOC-PES has an appropriate factor structure capturing all seven structures of the EDC in PE at the situational level, in line with Smith et al.’s [11] suggestions. Across all perceived climate dimensions of the MUMOC-PES a large amount of between-class variance emerged, suggesting that this new measure can capture variation in motivational climate due to different contextual factors such as teachers’ behaviors.

In line with hypothesis 2, the high positive correlations among empowering scales’ scores and the strong positive correlations among disempowering scales’ scores supported the internal convergent validity for the MUMOC-PES. On the other hand, the negative associations between empowering and disempowering scales’ scores supported the internal divergent validity for the MUMOC-PES.

The positive associations between PETEG mastery and MUMOC-PES task-involving scales and between PETEG performance and MUMOC-PES ego-involving scales, are also in agreement with hypothesis 2. These findings support the concurrent validity of the task- and ego-involving scales of the MUMOC-PES. The positive correlations between disempowering climate dimensions of MUMOC-PES and PETEG performance-approach/avoidance scales, as well as the positive relationships between MUMOC-PES empowering scales and PETEG mastery scale, support the external convergent validity of all MUMOC-PES climate scales. Finally, the negative associations between PETEG performance scales and MUMOC-PES empowering scales and between PETEG mastery scale and MUMOC-PES disempowering scales, provide evidence for external divergent validity for the MUMOC-PES. 

These results are in line with previous studies suggesting that perceived mastery climate, perceptions of autonomy and relatedness support in class correspond positively to learning and positive attitudes towards physical activity, while perceived performance climates are considered more controlling [39,55].

Notably, the present study unveiled the importance of including structure in EDC measures. As Sierens et al. [56] and Vansteenkiste et al. [39] suggested, structure describes the degree to which a social context is organized, and, at the same time, is the communication key component of the guidelines and expectations of the teachers or coaches. Teachers who provide a well-structured environment and clear rules create an efficient communicative context; their expectations are responsive to student needs, helpful, and supportive, deploying a learning environment [56,57]. When the structural elements are engaged in an autonomy-supportive way, the outcome is satisfaction, engagement and self-determined motivation [38]. Structure facilitates learning and intrinsic satisfaction in PE because in a well-structured class students find challenging tasks, receive helpful feedback, they can more easily become task-involved, adopt mastery goals and form intentions to behave adaptively in PE [5]. Our findings in PE stemming from the no mediation model are in line with Jang et al. [38] who found that autonomy support and structure positively predicted high school students’ behavioral engagement and satisfaction in the academic domain. Importantly, our second model, Figure 1, suggesting that the effects of perceived structure on satisfaction in PE were fully mediated through perception of mastery climate that clearly promotes mastery goal adoption, are in line with AGT that a structured environment helps students adopt mastery goals [4] and promote intrinsic satisfaction in PE.

All disempowering climate dimensions, particularly relatedness thwarting, were negatively associated with satisfaction in PE. Indeed, relatedness thwarting accounted for unique variance in satisfaction. These findings support the importance of capturing perceived relatedness thwarting through EDC measures, which did not happen in previous EDC studies. These negative effects might be both direct and indirect through diminishing perceptions of mastery climate and mastery goal adoption. Thwarting relationships with the teacher does not help students concentrate on learning and form intentions to learn. These findings are in line with past research implying that perceived climate that diminishes (thwarts) students-teacher relationship, negatively affects student satisfaction in PE classes [58,59]. 

Perceptions of need thwarting behaviors are also linked to other disempowering climate dimensions, such as perceptions of controlling environment that also results to maladaptive outcomes [58,60]. Hence, PE teachers should be cautious about teaching behaviors that might be controlling and thwart their relationships with students, because students are negatively affected by these behaviors, impacting their motivation and participation in PE classes [58].

The SEM analysis results support H_3_ implying that the effects of the MUMOC-PES’s thwarting and structure scales are more distal to motivational outcomes in PE than PETEG mastery scale, because the latter were constructed based on the rule of compatibility implying a perception-goal linkage. Moreover, the task-involving scale of the MUMOC-PES did not account for unique variance of satisfaction in PE. These findings imply that both the MUMOC-PES and the PETEG measures might be useful research tools, depending on the researchers’ goals. If one wants to capture perceived motivational climate associated with motivational outcomes, then the mastery and performance scales of the PETEG might be more preferable than the task-involving and ego-involving scales of the MUMOC-PES. On the other hand, the task and ego-involving scales of the MUMOC-PES might be more useful in research involving triangulation research methods (e.g., involving both perceived and observed climate) or in teacher training, because an external observer cannot easily observe whether class structures trigger student goal adoption. Of course, further research is needed to substantiate these arguments.

On the other hand, MUMOC-PES scales linked with SDT (autonomy and relatedness thwarting) had direct effects on satisfaction, which is in line with SDT that basic needs satisfaction and thwarting have direct impact on motivational outcomes beyond activation of achievement goals. Perception of autonomy plays a vital role in student satisfaction. The findings of previous studies show that the demonstration of autonomy-supportive climate could effectively improve the motivational climate in class and student competence and satisfaction [36,60].

The SEM findings revealed that estimating the motivational climate as mastery-oriented is adaptive [18,33,61]. Moreover, empowering climate positively predicted student satisfaction in PE. These results are in line with past research [8,44]. Finally, age and gender had a negative impact on satisfaction, suggesting that higher age and being a female is negatively linked with satisfaction in PE. These findings have emerged several times in past research [62,63].

In the literature of EDC as was conceptualized by Duda [3], this is one of the first studies showing through multilevel analysis that between-teacher variation in perceived empowering climate is significant positive predictor of student satisfaction in PE (H_4_). From a methodological standpoint, these are novel results compared to Miltons et al.’s [8] who did not report multilevel analyses. The sum of these findings adds to existing evidence about the critical role of teachers in creating mastery and empowering climates to promote student motivation in PE [3,8].

Between-teacher variation in perceived disempowering climate was not a predictor of student satisfaction. We used only one short measure of motivational outcomes capturing satisfaction due to time constraints in the completion of the questionnaire. If our DV was a negative motivational outcome, such as negative affect, the effects of between-teacher variation in perceived disempowering climate might have been significant. It might be worthy to examine this assumption in future research using the MUMOC-PES.

## 8. Implications

The present study’s findings have important recommendations for PE teachers and policy makers because they revealed that empowering dimensions of motivational climate (autonomy support, task-involving, relatedness support, structure) were positively related to pupil satisfaction. Both pre-service and in-service PE teachers should be trained to become more aware of particular behaviors and teacher-student communication that lead to an empowering climate and the development of a warm and caring class context that promotes pupil participation in the learning process [8,27]. The PE teachers should be also trained regarding what constitutes disempowering teaching behavior and how to prepare themselves to avoid disempowering behaviors particularly in challenging situations (e.g., coping with student misbehaviors). 

Policy makers and school principals should support and not thwart teachers’ psychological needs because this has an impact on teachers’ teaching and the empowering or disempowering climate that they create in their classes [16,64]. For example, a controlling prescriptive curriculum, or a work environment pressuring teachers or diminishing the value of the PE lesson and teachers’ job enjoyment, thwart PE teachers’ basic psychological needs, leading them to burnout and to ill-health [16,65], According to Deci and Ryan [13], the first stage in boosting motivation is to create a caring environment. A supportive, warm, and friendly work environment leads to a better perceived sense of autonomy when the demand for competence is satisfied rather than frustrated [13,66].

## 9. Limitations and Future Research

Although the MUMOC-PES was developed for concurrent use with the MMCOS [11], this research has yet to be reported. A positive association of MUMOC-PES and MMCOS subscales would provide evidence of validity for both measures. The representativeness of the samples deriving from one European country might be another limitation; future studies should examine its construct validity across different countries. Some MUMOC-PES subscales revealed low internal consistency. These subscales might need further improvement in the future. The present study focused on one motivational outcome, satisfaction. Future research should examine additional motivational outcomes, particularly the effects of empowering and disempowering subscales on maladaptive outcomes. 

## 10. Conclusions

The current study supports Duda’s [3] Empowering-Disempowering Climate conceptualization based on Achievement Goal Theory and Self-Determination Theory and provides a tool that might help researchers and practitioners extend this line of research and applications in Physical Education and youth sport settings. This tool can be adapted and further refined in research across countries, particularly in triangulation designs that might also adopt the Multidimensional Motivational Climate Observation System to capture both students’ and external observers’ perceptions. The two instruments might be useful in training PE teachers to improve their practices to maximize empowerment and minimize disempowerment of students in PE. 

## Figures and Tables

**Figure 1 ijerph-20-04202-f001:**
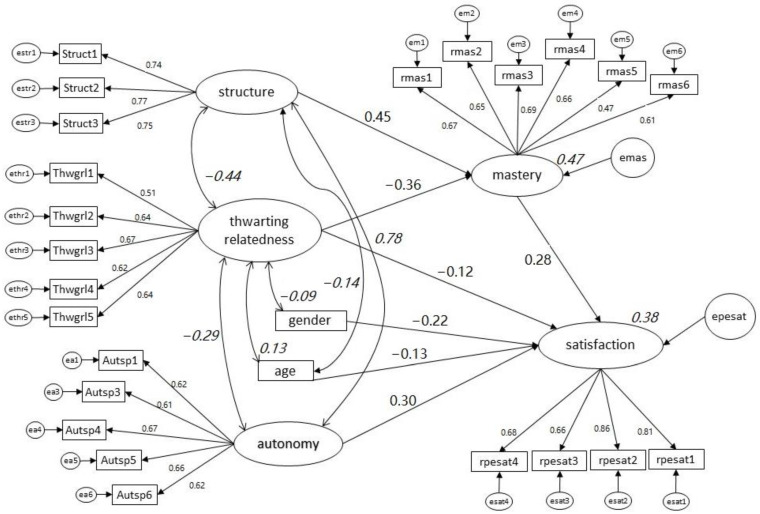
SEM model extracted from AMOS output investigating direct and indirect effects of perceived motivational climate dimensions, gender, and age on satisfaction in PE. Note: Coefficients in bold fonts in arrows indicate standardized beta weights and coefficients in italics fonts in curved lines indicate correlations. Only statistically significant (*p* < 0.001) standardized beta weights and correlations are shown (except of thwarting relationships → satisfaction, *p* < 0.01 and the correlation between gender and thwarting relationships, *p* < 0.02). Squared multiple correlation for mastery = 0.47 and for satisfaction = 0.38.

**Table 1 ijerph-20-04202-t001:** Factor loadings (standardized beta weights) stemming from CFA of the MUMOC-PES.

Factors							
Items	AS	CO	TI	EI	RS	RT	ST
1. acknowledged/took into consideration the perspectives of pupils	0.60						
2. emphasized interest/satisfaction for participating in an exercise or game	0.60						
3. justified/explained why we did the activities and why we needed to pay attention to something.	0.69						
4. provided opportunities to express our opinion about the activity we did	0.67						
5. encouraged the initiative of all pupils	0.62						
6. intimidated or threatened the pupils		0.58					
7. used expressions that were intended to control us and do what he/she wants		0.60					
8. was trying to impose us only what he/she wanted		0.68					
9. gave each of us information about how we performed an activity to improve			0.68				
10. explained the importance and the role of each of us in the activities			0.73				
11. emphasized and recognized our effort in order to improve			0.76				
12. divided us into cooperation groups			0.57				
13. encouraged the competition between us				0.52			
14. praised only the athletically capable pupils and downgraded the less capable				0.49			
15. punished us when we made a mistake				0.61			
16. spoke/communicated with us in a very warm way					0.60		
17. accepted all of us (tried to include all of us in all activities)					0.72		
18. showed interest and care for everyone (e.g., when someone was upset or hurt)					0.70		
19. was not interested, nor did he/she take care of us when we had a problem (e.g., when someone was upset)						0.52	
20. diminished us and did not recognize our efforts						0.65	
21. spoke/communicated with us in a negative (cold, critical) way						0.67	
22. excluded some pupils from some activities						0.64	
23. avoided talking to us						0.60	
24. gave instructions to organize better the class							0.75
25. was expecting us to learn something from the activities							0.77
26. guided and advised us throughout the lesson							0.74

Note: AS: Autonomy Support; CO: Controlling; TI: Task-Involving; EI: Ego-Involving; RS: Relatedness Support; Relatedness Thwarting; ST: Structure. For all standardized beta weights, *p* < 0.001.

**Table 2 ijerph-20-04202-t002:** Means (*M*), Standard Deviations, Cronbach’s *α*, McDonald’s omega (ω) Correlations among the examined variables and Effect Sizes (*ε*^2^) concerning between-Classes Differences.

Variables	*M*	*SD*	*α*	*ω*	*ε^2^*	Correlations											
						1	2	3	4	5	6	7	8	9	10	11	12
MUMOC-PES																	
1. Autonomy support	2.00	0.69	0.77	0.78	0.36	*-*											
2. Task-involving	2.10	0.74	0.77	0.77	0.40	0.68 **	-										
3. Relatedness Support	2.40	0.60	0.70	0.71	0.27	0.44 **	0.51 **	-									
4. Structure	2.22	0.78	0.80	0.80	0.38	0.61 **	0.70 **	0.56 **	-								
5. Controlling	0.62	0.68	0.65	0.67	0.19	−0.30 **	−0.24 **	−0.28 **	−0.24 **	-							
6. Ego-involving	0.68	0.69	0.56	0.56	0.16	−0.12 **	−0.14 **	−0.25 **	−0.13 **	0.45 **	-						
7. Relatedness Thwarting	0.48	0.62	0.75	0.75	0.23	−0.23 **	−0.30 **	−0.52 **	−0.34 **	0.45 **	0.43 **	-					
PETEG																	
8. Mastery	4.10	0.74	0.79	0.79	0.29	0.37 **	0.47 **	0.52 **	0.50 **	−0.19 **	−0.13 **	−0.43 **	-				
9. Performance approach	2.71	0.83	0.67	0.67	0.11	−0.10 **	−0.11 **	−0.19 **	−0.10 **	0.15 **	0.31 **	0.24 **	−0.15 **	-			
10. Performance avoidance	2.38	0.80	0.60	0.60	0.14	−0.20 **	−0.23 **	−0.25 **	−0.20 **	0.32 **	0.35 **	0.36 **	−0.23 **	0.43 **	-		
PUPIL SATISFACTION																	
11. Satisfaction in PE	4.10	0.82	0.86	0.85	0.12	0.38 **	0.37 **	0.36 **	0.40 **	−0.14 **	−0.08 *	−0.27 **	0.40 **	−0.07 *	−0.14 **	-	
DEMOGRAPHIC																	
12. Gender	-	-	-	-	-	−0.03	0.04	−0.00	0.03	−0.11 **	−0.12 **	−0.09 **	0.04	−0.04	−0.10 **	−0.21 ***	-
13. Age	13.87	1.76	-	-	-	−0.06	−0.20 **	−0.18 **	−0.16 **	−0.11 **	−0.01	0.12 **	−0.15 **	0.00	0.03	−0.20 **	−0.02

*** *p* < 0.001, ** *p* < 0.01, * *p* < 0.05; for all η^2^, *p* < 0.001.

**Table 3 ijerph-20-04202-t003:** Estimates of Fixed Effects from Multilevel Regression Analysis with Satisfaction as Dependent Variable.

	Est.	S.E.	t
Intercept	4.57	0.40	11.53 ***
Gender	−0.39	0.04	−8.75 ***
Age	−0.06	0.02	−3.46 ***
Empowering-I	0.63	0.05	12.01 ***
Disempowering-I	−0.23	0.05	−4.45 ***
Empowering-C	0.44	0.09	4.88 ***
Disempowering-C	0.09	0.13	0.71

Note: I = Individual perceptions within class, centered on the class Mean (level 1 predictor); C = Class mean for climate perception (level 2 predictor); Empowering = (average score on all empowering climate scores: Autonomy, Task-involving, Relatedness supporting, Structure); Disempowering = (average score on all disempowering climate scores: Controlling, Ego-involving and Relatedness thwarting climate), Est. = Estimate of Fixed Effects; S.E. = Standard Error. **** p* < 0.001.

## Data Availability

The data used to support the findings of this study are available from the authors upon request.

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
