# Peer review of "Multidimensional Motivational Climate Questionnaire in Physical Education at the Situational Level of Generality (MUMOC-PES)"

_ijerph, 2023, doi:10.3390/ijerph20054202_

Round 1

Reviewer 1 Report

I congratulate the authors for the work carried out and the contribution made to the scientific literatura.

I think the authors should review this part of the abstract: capturing four dimensions of empowering (autonomy support, task-involvement, relatedness support, structure) and three dimensions of disempowering (controlling, relatedness thwarting) climate. Missing a dimension in disempowering climate?

PAGE 3 LINE 110 - I don't know if I have understood well what the authors say: "This instrument captures three dimensions of 110 task-involving climate (teacher-initiated learning orientation and students' learning ori-111 entation) and two dimensions of ego-involving climate (students' competitive orientation, 112 students' worries about mistakes and outcome without effort) [29]."

I think it should be written this way: This instrument captures TWO dimensions of 110 task-involving climate (teacher-initiated learning orientation and students' learning ori-111 entation) and THREE dimensions of ego-involving climate (students' competitive orientation, 112 students' worries about mistakes and outcome without effort) 

PAGE 5 LINE 222 - In Europe there are very diverse cultures. I think they should mention the country where the data was collected.

PAGE 7 LINE 323 - The reliability alpha coefficients for the scales comprising the remaining items of each 323 aforementioned factor were aceptable (.56 to .84). Authors must justify why they accept data below .70. Different scientific papers accept them, but the authors must justify it.

I recommend that authors control the SEM by educational level, since there are three educational levels.

In addition, I recommend an invarainza by level of studies.

Author Response

Manuscript ID: IJERPH-2184115

Response letter

Dear Editor

Thank you and your reviewers for allowing us to submit a revised draft of the manuscript “Multidimensional Motivational Climate questionnaire in Physical Education at the Situational level of generality (MUMOC-PES)” for publication in the International Journal of Environmental Research and Public Health. We appreciate the time and effort that you and your reviewers dedicated to providing feedback on our manuscript and we are grateful for the helpful comments and valuable suggestions for our paper. We have incorporated your recommendations. Those changes are highlighted within the manuscript. Please see below point-by-point the responses to your comments and related references to page and line of the revised manuscript.

Reviewer 1

Reviewers' Comments to the Authors:

  1. I think the authors should review this part of the abstract: capturing four dimensions of empowering (autonomy support, task-involvement, relatedness support, structure) and three dimensions of disempowering (controlling, relatedness thwarting) climate. Missing a dimension in disempowering climate?

Authors’ response:

  • Thank you, please see revision in abstract page 2, line 6, please see the addition of the missing dimension “ego-involvement” in disempowering climate.

Reviewers' Comments to the Authors:

  1. PAGE 3 LINE 110 - I don't know if I have understood well what the authors say: "This instrument captures three dimensions of task-involving climate (teacher-initiated learning orientation and students' learning orientation) and two dimensions of ego-involving climate (students' competitive orientation, students' worries about mistakes and outcome without effort) [29]."

I think it should be written this way: This instrument captures TWO dimensions of task-involving climate (teacher-initiated learning orientation and students' learning orientation) and THREE dimensions of ego-involving climate (students' competitive orientation, students' worries about mistakes and outcome without effort).

Authors’ response:

  • Thank you! it was corrected as you suggested (page 5, lines 19, 20 and 21).

Reviewers' Comments to the Authors:

  1. In Europe there are very diverse cultures. I think they should mention the country where the data was collected.

Authors’ reponse:

  • In the revision we added that the country was Greece (page 8, line 30).

Reviewers' Comments to the Authors:

  1. The reliability alpha coefficients for the scales comprising the remaining items of each aforementioned factor were acceptable (.56 to .84). Authors must justify why they accept data below .70. Different scientific papers accept them, but the authors must justify it.

Authors’ response:

  • Please see revision (page 12, lines 4-8). Please also see our note and suggestion in the limitations section (Page 18, Lines 16-17).

Reviewers' Comments to the Authors:

  1. I recommend that authors control the SEM by educational level, since there are three educational levels.

Authors’ response:

  • Thank you for the suggestion. Educational level takes three values capturing three different periods of students’ age: (1) elementary school (students’ age 10-11 in our sample), (2) jounior high school (ages 12-14), (3) senior high school (ages 15-18).  Taking into consideration that we have already controlled the SEM by age, addition of educational level into the SEM would not add something new, because the effects of age are similar to the effects of educational level and probably stronger/more robust than educational level, because age is a continuous variable while educational level is discrete variable.

Reviewer 2 Report

The MS presents the validation of the Motivational Climate questionnaire, together with an assessment of the relationships with mastery, satisfaction, age and gender.

It is overall well-written and structured as well informative. Hence, I would suggest publication pending the revisions listed below.

1.       Introduction. It is rich and informative. However I would stress more that teacher enthusiasm (e.g., Keller et al., 2016; Moè & Katz, 2022) and adoption of need supportive practices (e.g., Aelteman et al., 2019; Moè et al., 2022) favor a range of student outcomes including learning (e.g., De Beni & Moè, 2003; Kunter et al., 2011; Moè et al., 2021)

Suggested references

Aelterman, N., Vansteenkiste, M., Haerens, L., Soenens, B., Fontaine, J. R., & Reeve, J. (2019). Toward an integrative and fine-grained insight in motivating and demotivating teaching styles: The merits of a circumplex approach. Journal of Educational Psychology, 111(3), 497-521.

Carraro, A., Gobbi, E., & Moè, A. (2017). More gyms or more psychological support? Preventing burnout and supporting job satisfaction in physical education teachers. Sport Sciences for Health, 13(1), 55-62.

De Beni, R., & Moè, A. (2003). Imagery and rehearsal as study strategies for written or orally presented passages. Psychonomic Bulletin & Review, 10(4), 975-980.

Keller, M. M., Hoy, A. W., Goetz, T., & Frenzel, A. C. (2016). Teacher enthusiasm: Reviewing and redefining a complex construct. Educational Psychology Review, 28(4), 743-769.

Kunter, M., Frenzel, A., Nagy, G., Baumert, J., & Pekrun, R. (2011). Teacher enthusiasm: Dimensionality and context specificity. Contemporary Educational Psychology, 36(4), 289-301.

Moè, A., Consiglio, P., & Katz, I. (2022). Exploring the circumplex model of motivating and demotivating teaching styles: The role of teacher need satisfaction and need frustration. Teaching and Teacher Education, 118, 103823.

Moè, A., Frenzel, A. C., Au, L., & Taxer, J. L. (2021). Displayed enthusiasm attracts attention and improves recall. British Journal of Educational Psychology, 91(3), 911-927.

Moè, A., & Katz, I. (2022). Need satisfied teachers adopt a motivating style: The mediation of teacher enthusiasm. Learning and Individual Differences, 99, 102203.

2.       Within the SDT studies typically they distinguish between a dark and a bright side. Consequently it is not clear why in the model relatedness thwarting instead of related support was included

3.       Discussion. I would create a standing alone section Educational Implications by also including something about the factors leading teachers to be less supportive, e.g. Carraro et al., 2017. Then, I would also write something about generalizability of the results to school subjects different from PE

4.       Overall. There are plenty of acronyms which I think make the reading not fluent: it is necessary every time to go back…maybe reduce them

I wish the AA the best with their research!

Author Response

Manuscript ID: IJERPH-2184115

Response letter

Dear Editor

Thank you and your reviewers for allowing us to submit a revised draft of the manuscript “Multidimensional Motivational Climate questionnaire in Physical Education at the Situational level of generality (MUMOC-PES)” for publication in the International Journal of Environmental Research and Public Health. We appreciate the time and effort that you and your reviewers dedicated to providing feedback on our manuscript and we are grateful for the helpful comments and valuable suggestions for our paper. We have incorporated your recommendations. Those changes are highlighted within the manuscript. Please see below point-by-point the responses to your comments and related references to page and line of the revised manuscript.

Reviewer 2

Comments to the Authors:

Introduction. It is rich and informative. However I would stress more that teacher enthusiasm (e.g., Keller et al., 2016; Moè & Katz, 2022) and adoption of need supportive practices (e.g., Aelteman et al., 2019; Moè et al., 2022) favor a range of student outcomes including learning (e.g., De Beni & Moè, 2003; Kunter et al., 2011; Moè et al., 2021)

Authors’ response:

In page 3, lines 28-29, you’ll find addition of the following references:

Aelterman, N.; Vansteenkiste, M.; Haerens, L.; Soenens, B.; Fontaine, J. R. J.; Reeve, J. Toward an Integrative and Fine-Grained Insight in Motivating and Demotivating Teaching Styles: The Merits of a Circumplex Approach. Journal of Educational Psychology, 2019, 111(3), 497–521.

Carraro, A.; Gobbi, E.; Moè, A. More gyms or more psychological support? Preventing burnout and supporting job satisfaction in physical education teachers. Sport Sciences for Health, 2017, 13(1), 55-62.

Moè, A.; Frenzel, A. C.; Au, L.; Taxer, J. L. Displayed enthusiasm attracts attention and improves recall. British Journal of Educational Psychology, 2021, 91(3), 911-927.

Reviewers' Comments to the Authors:

Within the SDT studies typically they distinguish between a dark and a bright side. Consequently it is not clear why in the model relatedness thwarting instead of related support was included

Authors’ response:

  • As we explained in the results (Page 13, Lines 16-18), in the initial model that we tested, while relatedeness thwarting contributed significantly in the model, relatedness support did not contribute significantly. Thus, as we explained in the results, in the final model (presented in Figure 1), only relatedness thwarting was retained.

Reviewers' Comments to the Authors:

Discussion. I would create a standing alone section Educational Implications by also including something about the factors leading teachers to be less supportive, e.g. Carraro et al., 2017. Then, I would also write something about generalizability of the results to school subjects different from PE

Authors’ response:

  • Thank you for the comment, please see the new paragraph (page 17, lines 20-22 and 27-29, and page 18, lines 1-11).

Reviewers' Comments to the Authors:

 Overall. There are plenty of acronyms which I think make the reading not fluent: it is necessary every time to go back…maybe reduce them.

Authors’ response:

We generally agree, but most of the acronyms refer to instruments, hence they are used as labels;  meaning might not be properly conveyed if some acronyms are removed from the text.

Sincerely,